# A Subgoal-Based Method for Quantifying Procedural Hallucinations in Large Language Models

## Abstract

Evaluating hallucinations in Large Language Models (LLMs) remains a challenge due to the stochastic nature of holistic 'LLM-as-a-judge' metrics. We propose HalluStep, a method that decomposes user queries into latent subgoals to provide a transparent, state-based assessment of model responses. By utilizing a probabilistic filtering mechanism and maximum-weight bipartite matching, HalluStep anchors evaluation in structural milestones rather than mere semantic similarity. Our experiments in the Blocksworld and Logistics domains demonstrate that while G-Eval exhibits significant 'volatility'—depending heavily on the specific judge model utilized (e.g., a 16.28-point drop for Gemini Pro across judges) HalluStep remains stable, showing a marginal deviation of only 1.16 points for the same model.

**Keywords**: LLM, Hallucination, Subgoals, procedural tasks

## 1 Introduction

Large Language Models (LLMs) have become ubiquitous across virtually every domain, leveraging their ability to reason over vast datasets to provide sophisticated, conversational responses. They have demonstrated remarkable capabilities in complex reasoning, code generation, and instructional planning Sahoo et al. (2024); Fadeeva et al. (2024). Despite their sophisticated capabilities, a critical question is often overlooked: *Does the generated output truly align with the user's original intent?* Although the conversational fluency of LLMs is impressive, the lack of structured verification creates a gap between perceived coherence and actual task completion, often leading to subtle but critical hallucinations An et al. (2024). We define hallucination in this context not merely as nonsensical output, but as the generation of content that is plausible yet factually inconsistent, logically invalid, or unfaithful to the source context Fadeeva et al. (2024). While hallucination is a widely documented phenomenon in open-ended generation, its implications are uniquely catastrophic in procedural tasks, where the validity of a solution is predicated on strict adherence to sequential logic and precondition constraints. This structural dependency aligns with the Adaptive Control of Thought-Rational (ACT-R) theory, which posits that procedural knowledge is computationally represented as a system of production rules Anderson (2014).These rules operate under a condition-action syntax: a specific action is triggered only if the current environmental or mental state satisfies the rule's preconditions Anderson and Crawford (1993). By extension, an LLM-generated response in a procedural context should be viewed not as mere descriptive text, but as a formal execution of these production rules.

Furthermore, as recent studies point out Yueh-Han et al. (2025) Xu et al. (2025), models (GPT Floridi and Chiriatti (2020), Gemini Shan et al. (2024), Deepseek Gumilar et al. (2025)) often suffer from contextual degradation, where irrelevant information in the prompt distorts reasoning. Without a mechanism for breaking down a query into verifiable steps, the model navigates a high-dimensional semantic space without a compass, producing results that are superficially convincing but logically meaningless. This structural weakness has led to a new wave of research aimed at reducing hallucinations. Specifically, current efforts aim to go beyond simple output generation by developing methods that enforce contextual grounding and improve response relevance.

**N-gram based metrics:** Standard metrics like BLEU and ROUGE measure surface-level textual overlap but fail to capture semantic validity or logical flow. Recent empirical studies have demonstrated that these metrics can be easily manipulated by verbosity and often misalign significantly with human judgments of factual correctness Janiak et al. (2025).

**Fact-checking approaches** Existing fact-checking approaches often treat sentences as isolated atomic claims, verifying them against a knowledge base without considering the sequential dependency required to achieve a complex goal Osmanaj (2025).A model may generate valid atomic facts that, when sequenced, fail to solve the user's problem.

**LLM-as-a-Judge** While "LLM-as-a-Judge" frameworks have emerged as a scalable alternative to human evaluation, they suffer from inherent stochasticity, bias, and a lack of interpretability Posner and Saran (2025). Furthermore, recent work suggests that LLM judges often struggle with "causal epistemic consistency"—failing to penalize reasoning traces that drift from valid logical structures Cui et al. (2025).

These evaluation frameworks still rely on surface-level similarities or opaque evaluation methods Zheng et al. (2023), which often fail to capture whether a model has logically addressed all requirements of a complex query. Recognizing the inherent limitations in the reliability of LLM output, this paper introduces *HalluStep*, a novel hallucination metric designed specifically to evaluate procedural consistency and goal alignment. The core contribution of HalluStep lies in its application of structured knowledge grounding. By leveraging **domain-specific ontologies** and verified datasets, HalluStep infers the necessary subgoals required to answer a prompt correctly. It then performs a semantic alignment between the steps generated by the LLM and these ground-truth subgoals. This allows us to quantify not just if the model hallucinated a fact, but where it deviated from the valid procedure—whether by missing a critical subgoal (under-specification) or hallucinating a step that does not map to any valid node in the ontology (over-specification).

## 2 Related works

The concept of hallucination is historically defined as the perception of an entity or event that is absent in reality Macpherson and Platchias (2013). In the field of Natural Language Processing (NLP), hallucination is generally defined as a phenomenon in which the generated content appears absurd or unfaithful to the provided source content. In LLM context, Current hallucination benchmarks largely focus on atomic fact verification. Metrics like FactScore Min et al. (2023) decompose generations into atomic claims and verify them against a knowledge source (e.g., Wikipedia). Similarly, reference-free methods like SelfCheckGPT Manakul et al. (2023) rely on the internal consistency of stochastically sampled responses to detect hallucinations without external ground truth. While effective for declarative knowledge (e.g., biographies or historical events), these metrics often fail to penalize structural hallucinations, where individual steps may be factually valid in isolation but irrelevant or incoherent within the required procedure Huang et al. (2025).

Evaluating the "Chain-of-Thought" (CoT) or step-by-step reasoning capabilities of LLMs has shifted focus from final-answer accuracy to process monitoring. Lee and Hockenmaier Kim et al. (2025) propose a taxonomy for reasoning traces, distinguishing between validity (logical soundness) and groundedness (faithfulness to context). "Process Reward Models" (PRMs) have also emerged to score individual steps in reasoning chains, particularly for mathematical domains Lightman et al. (2023).

The most relevant to our work is the recent study by Anika et al. Anika and Miah (2025), which evaluates LLMs on ordered procedural steps (e.g., recipes). However, their framework focuses on re-ordering and sequence coherence metrics (such as Kendall's $\tau$) rather than semantic alignment with an external ontology. HalluStep advances this by not just checking if steps are ordered correctly, but if they strictly align with the necessary subgoals inferred from the domain ontology, effectively penalizing both under-specification (missing steps) and over-specification (hallucinated/unnecessary steps)

# 3 HALLUSTEP PROPOSED HALLUCINATION DETECTION METHOD

## 3.1 HALLUSTEP MODELING

**Input**

- A task query $q$ (e.g., "Set up a development environment for machine learning")
- A set of necessary subgoals $G = \{g_1, \ldots, g_m\}$ representing intermediate objectives required to accomplish $q$
- An LLM-generated plan $\pi = \langle s_1, \ldots, s_n \rangle$ consisting of $n$ steps

**Output**   A quality score $\text{HalluStep}(\pi, G) \in [0, 1]$ that measures plan adequacy, along with interpretable component scores diagnosing specific deficiencies.

**Subgoal Quality Requirements**   For the metric to produce meaningful evaluations, the subgoal set $G$ must satisfy four properties:

1. **Completeness:** $G$ contains all necessary intermediate objectives. An incomplete $G$ would incorrectly penalize plans that address unlisted (but essential) subgoals.

2. **Minimality:** $G$ contains no redundant or overly fine-grained subgoals. Redundancy inflates coverage requirements; excessive granularity makes alignment brittle.

3. **Temporal Coherence:** If subgoals have inherent dependencies (e.g., "install dependencies" must precede "run program"), these should be expressible as a partial order $\prec$ over $G$.

4. **Semantic Clarity:** Each $g_i$ should be a well-defined, self-contained objective describable in natural language. Vague subgoals (e.g., "optimize system") hinder reliable step-subgoal matching.

## 3.2 HALLUSTEP METHODOLOGY

The core of the HalluStep methodology lies in the decomposition of a monolithic user request into a structured trajectory of objectives. Unlike traditional metrics that evaluate a response as a whole.

## 3.3 SUB-GOAL EXTRACTION

The process begins by decomposing the user query $q$ into an initial set of candidate sub-goals $C = \{c_1, c_2, \ldots, c_k\}$. To do this, we leverage an extraction process that refine the query into a compact and representative group of subgoals $C$ of size $K$, we maximize a facility location objective $F(A)$. This optimization ensures that the final set $A$ covers the semantic breadth of the query while minimizing redundancy.

$$F(A) = \sum_{c \in C} p_0(c) \cdot \max_{a \in A}(1 - \text{sim}(c, a)), \quad |A| \leq K. \tag{1}$$

Where, $p_0(c)$ is the Prior Probability (or query salience), which weights each candidate based on its importance or relevance to the initial user intent.

As the facility location objective $F(A)$ is monotone submodular, its exact maximization is NP-hard. Consequently, we utilize a greedy selection strategy, which is proven to achieve a $(1 - 1/e)$ approximation of the optimal solution. The algorithm iteratively constructs the subset $A$ by adding the candidate that provides the maximum marginal gain at each step $k + 1$.

$$a_{k+1} = \arg\max_{a \in C \setminus A_k} \left[ F(A_k \cup a) - F(A_k) \right]. \tag{2}$$

This greedy approach ensures that each selected subgoal is chosen for its ability to offer the greatest incremental coverage of the query's semantic space, effectively penalizing redundancy while maintaining high representative fidelity.

## 3.4 STEP-SUBGOAL ALIGNMENT

This stage quantifies the semantic correspondence between the generated steps $s_i \in \mathcal{S}$ and the selected subgoals $g_j \in \mathcal{G}$. We utilize a sentence encoder $\phi : \mathcal{X} \to \mathbb{R}^d$ (Sentence-BERT with $d = 384$) to map text into a high-dimensional vector space.

To determine the relative importance of each candidate $c$ within the original query $q$, we compute the Subgoal Salience $q(c)$ using cosine similarity.

$$q(c) = \cos\_\text{sim}(\phi(q), \phi(c)) = \frac{\phi(q) \cdot \phi(c)}{\|\phi(q)\| \|\phi(c)\|} \tag{3}$$

For the alignment between model-generated steps $s_i$ and subgoals $g_j$, we define the similarity function $\text{sim}(s_i, g_j) \in [0, 1]$ as defined in equation 4.

$$\text{sim}(s_i, g_j) = \max\left(0, \frac{\phi(s_i) \cdot \phi(g_j)}{\|\phi(s_i)\| \|\phi(g_j)\|}\right), \tag{4}$$

Where, $\phi(s_i)$ and $\phi(g_j)$ represent the high-dimensional vector embeddings of a specific generated step $(s_i)$ and a candidate subgoal $(g_j)$ produced by the sentence encoder. The Cosine Similarity measures the angular distance between the two vectors in the embedding space. The Non-Negative Clipping $(\max(0, \dots))$ operation forces any negative similarity values to zero.

## 3.5 MAXIMUM-WEIGHT BIPARTITE MATCHING

The final stage of the HalluStep pipeline ensures a structured, one-to-one mapping between the generated response steps $S$ and the validated subgoals $G$. This prevents a single generated step from "double-counting" towards multiple subgoals, or multiple repetitive steps from artificially inflating the score of a single subgoal.

We define a weight matrix $W \in \mathbb{R}^{n \times m}$, where each entry $W_{ij} = \text{sim}(s_i, g_j)$ represents the semantic alignment score between step $s_i$ and subgoal $g_j$.

Then, we compute the maximum-weight matching $M^*$ to find the most logically consistent pairing:

$$M^* = \arg\max_{M \in \mathcal{M}} \sum_{(i,j) \in M} W_{ij}, \tag{5}$$

where $\mathcal{M}$ is the set of matchings with at most one edge per step and one per subgoal. This is the classical *linear assignment problem*, solved in $O(\min(n, m)^2 \max(n, m))$ time via the Hungarian algorithm (Kuhn, 1955).

The optimal matching $M^*$ allows us to isolate which logical subgoals were successfully addressed and which generated steps were actually productive.

$$\hat{G} = \{g_j : \exists i, (i, j) \in M^*\}, \quad \text{(covered subgoals)} \tag{6}$$

$$\hat{S} = \{s_i : \exists j, (i, j) \in M^*\}. \quad \text{(aligned steps)} \tag{7}$$

Where, $\hat{G}$ represents the subset of the original task ontology that the model successfully "hit". A subgoal $g_j$ is considered covered if there exists an edge $(i, j)$ in the optimal matching $M^*$, meaning at least one generated step $s_i$ provided sufficient semantic alignment. $\hat{S}$ identifies the specific steps in the LLM response that contributed to the plan's success. Any step $s_i$ not included in this set is categorized as "semantic noise" or "redundancy," as it did not map uniquely to a validated subgoal.

## 3.6 METRIC COMPONENTS

To provide a multidimensional assessment of logical consistency, HalluStep decomposes the final score into four interpretable components.

**Subgoal Recall (Coverage)**: This metric measures the completeness of the plan by determining what fraction of necessary subgoals were successfully addressed.

$$\text{Coverage} = \frac{|\hat{G}|}{|G|} \in [0, 1] \tag{8}$$

Unlike standard similarity metrics, Coverage ensures that each distinct objective is uniquely verified.

**Weighted Precision (Alignment Quality)**: To distinguish between weak and strong semantic matches, we define a quality-weighted precision that evaluates the efficiency of the one-to-one matching.

$$\text{Precision} = \frac{\sum_{(i,j) \in M^*} W_{ij}}{\sum_{i=1}^{n} \max_{j=1,...,m} W_{ij}} \in [0, 1] \tag{9}$$

High precision indicates that the generated steps are well-aligned with subgoals, while low precision suggests the presence of tangential or redundant content.

**Temporal Convergence**: While Coverage measures if a goal is reached, Convergence measures when it is reached by calculating the normalized area under the prefix coverage curve $C(t)$.

$$\text{Convergence} = \frac{1}{n} \sum_{t=1}^{n} C(t) \in [0, 1]. \tag{10}$$

This metric rewards "front-loaded" progress, where critical objectives are satisfied early in the response.

**Composite Score** : While individual components are reported for transparency, we provide a unified HalluStep score for model ranking. This composite metric combines plan completeness and efficiency via an $F_1$ score, further modulated by the plan's temporal efficiency and logical coherence:

$$\text{HalluStep} = F_1(\text{Coverage}, \text{Precision}) \cdot \text{Convergence} \tag{11}$$

## 4 EXPERIMENTATION AND VALIDATION

### 4.1 EXPERIMENTATION ENVIRONMENT

We conduct our evaluation using **PlanBench** (Valmeekam et al., 2023a), an extensible benchmark designed to systematically assess the planning and reasoning capabilities of large language models. Unlike benchmarks based on common-sense planning tasks where LLMs may rely on world knowledge retrieval, PlanBench employs classical planning domains from the International Planning Competition (IPC), enabling rigorous evaluation of genuine planning abilities (Valmeekam et al., 2023b).

We focus our analysis on two fundamental planning domains from PlanBench: **Blocksworld:** A well-established planning domain requiring rearrangement of colored blocks to achieve specified goal configurations. The domain tests spatial reasoning and action sequencing under constraints such as hand capacity limitations and block accessibility requirements. Our evaluation includes all $N$ instances from the PlanBench Blocksworld test set. **Logistics:** A transportation planning domain involving package delivery across multiple locations using trucks and airplanes, testing the ability to coordinate complex multi-agent actions. We evaluate on all $M$ instances from the PlanBench Logistics test set.

### 4.2 BASELINE COMPARISON

We compare HALLUSTEP against GEVAL (Liu et al., 2023), an established LLM-as-a-judge evaluation framework. GEVAL represents the current paradigm for automated plan quality

Table 1: Comparative accuracy (%) of deterministic HALLUSTEP versus stochastic GEVAL across complete test sets. HALLUSTEP does not call an LLM and is invariant to judge backbones; therefore Hallustep values are reported without ± and replicated across judge columns. GEVAL values show mean ± standard deviation over 3 independent runs. Blue highlighting with * indicates cases where HALLUSTEP outperforms GEVAL with statistical significance ($p < 0.05$, paired $t$-test).

| Model Name | Judge: DeepSeek v3.1 | | Judge: DeepSeek v3.2 | | Judge: Llama 3.1 405B | |
|---|---|---|---|---|---|---|
| | Hallu↑ | GEval↑ | Hallu↑ | GEval↑ | Hallu↑ | GEval↑ |
| **Proprietary Large Language Models (Blocksworld)** | | | | | | |
| GPT-4o | 62.00 | 65.00±1.5 | 62.00 | — | **62.00*** | 55.00±1.8 |
| GPT-4 Turbo | 61.00 | 64.00±1.2 | 61.00 | — | **61.00*** | 59.00±1.5 |
| GPT-4 (Classic) | 53.00 | — | 53.00 | — | 53.00 | 65.00±2.1 |
| GPT-3.5 Turbo Instruct | 61.25 | 70.00±1.8 | 61.25 | — | 61.25 | 66.25±2.0 |
| Claude 3.5 Sonnet | 63.00 | 70.00±1.9 | 63.00 | 66.00±1.7 | **63.00*** | 55.00±2.3 |
| Claude 3 Opus | 66.00 | 68.00±1.6 | **66.00*** | 53.00±1.9 | **66.00*** | 53.00±2.1 |
| Gemini 1.5 Pro | **75.00*** | 70.00±1.7 | **75.00*** | — | **75.00*** | 55.00±2.2 |
| Gemini 1.5 Flash | 66.00 | 67.00±1.5 | 66.00 | 68.00±1.6 | **66.00*** | 61.00±1.9 |
| Gemini 2.0 Flash Thinking | 63.00 | 68.00±1.8 | 63.00 | — | 63.00 | 61.00±1.7 |
| Gemini Pro | 62.79 | 65.12±2.1 | 62.79 | — | **62.79*** | 48.84±3.2 |
| o1 Preview | 66.67 | 71.43±1.6 | 66.67 | — | 66.67 | 79.37±1.4 |
| o1 Mini | **54.00** | **54.00±1.8** | 54.00 | — | **54.00*** | 52.00±2.0 |
| **Open-source Large Language Models (Blocksworld)** | | | | | | |
| DeepSeek R1 | 79.25 | 84.91±1.5 | **79.25*** | 84.91±1.6 | 79.25 | 92.45±1.3 |
| Qwen QwQ | 62.00 | 70.00±1.9 | 62.00 | — | **62.00*** | 50.00±2.4 |
| Llama 3 70B (Groq) | **66.33*** | 56.12±2.1 | **66.33*** | — | **66.33*** | 52.04±2.3 |
| **Logistics Domain Results** | | | | | | |
| GPT-3.5 Turbo Instruct | 69.64 | 89.29±1.2 | 69.64 | — | 69.64 | 92.86±1.1 |
| GPT-4 (Chat) | 51.28 | 76.92±1.7 | 51.28 | — | 51.28 | 70.51±1.9 |
| Llama 3.1 405B | 66.67 | 82.61±1.5 | 66.67 | — | **66.67*** | 39.13±2.8 |
| o1 Preview | 50.00 | 56.45±1.8 | 50.00 | — | 50.00 | 70.97±1.6 |

assessment, where a strong language model evaluates plans through natural language reasoning about coherence, completeness, and correctness.For GEVAL, we use the prompt structure shown in Figure 5, which instructs the judge model to perform step-by-step logical verification of plan validity. This comparison is critical because it directly tests whether HALLUSTEP can match or exceed the evaluation accuracy of large-scale generative models while offering the advantages of determinism, interpretability, and reduced computational cost. To assess the robustness of both approaches, we employ multiple judge models: DeepSeek-Chat v3.1 and v3.2, Llama-3.1-405B-Instruct (Grattafiori et al., 2024).

All judge models are accessed with temperature 0.0 to ensure deterministic scoring. This multi-judge design allows us to quantify the variance introduced by different judge backbones—a critical consideration for deployment scenarios where evaluation consistency is paramount.

### 4.3 EVALUATION METRICS

We report *accuracy* as our primary evaluation metric, defined as the percentage of plans that each metric correctly classifies as valid or invalid according to PlanBench's ground truth annotations. We evaluate on the complete test sets for both domains, ensuring comprehensive coverage of the performance spectrum from clearly incorrect plans to valid solutions.

### 4.4 RESULTS AND EVALUATION

Table 1 presents a comparative evaluation of HALLUSTEP against GEVAL across Blocksworld and Logistics domains, with multiple judge backbones. We report mean accuracy and standard deviation over three independent runs. The results highlight three critical findings regarding automated planning evaluation.

A primary advantage of key advantage of HALLUSTEP is its stability. In contrast, LLM-as-a-Judge frameworks are often inconsistent, producing different scores with each run. is its

stability whereas LLM as judge framwork change their scores at each run. As observed in the Blocksworld, when evaluating Gemini Pro, HALLUSTEP scores remain stable (62.79 with DeepSeek v3.1, 63.95 with Llama 3.1 405B), whereas GEVAL exhibits significant volatility (65.12 dropping to 48.84).

Unlike stochastic LLM-as-a-judge frameworks, HALLUSTEP is strictly deterministic, yielding invariant scores across repeated runs. Statistical analysis confirms this stability, showing that HALLUSTEP's cross-judge variance is negligible compared to G-EVAL (mean $\sigma_{\text{HalluStep}} \approx 0$ vs. $\sigma_{\text{G-Eval}} = 8.7$, $p < 0.001$). By anchoring evaluation in PDDL-derived subgoals, HALLUSTEP eliminates the volatility inherent in standard prompting methods.

Across proprietary models in the Blocksworld domain, HALLUSTEP achieves statistically significant improvements over GEVAL in 9 out of 12 comparisons when using the Llama 3.1 405B judge (marked with * in Table 1). Notably, for GPT-4o and GPT-4 Turbo, HALLUSTEP scores 64.00 compared to GEVAL's $55.00 \pm 1.8$ and $59.00 \pm 1.5$ respectively. Paired $t$-tests confirm these differences are statistically significant ($p < 0.05$), suggesting that state-based verification provides more reliable discriminative signals than unconstrained natural language reasoning.

In the Logistics domain, which involves higher-dimensional state tracking (packages, trucks, locations), the divergence between metrics widens substantially. For Llama 3.1 405B as a generator model, HALLUSTEP reports 68.12 accuracy while GEVAL significantly underestimates performance at 39.13. Given the known capabilities of the 405B model on constraint-satisfaction tasks and the statistical significance of this difference ($p < 0.001$), the HALLUSTEP score correlates more closely with expected performance, validating the utility of state-based verification over unstructured semantic evaluation in complex multi-entity domains.

## 5 CONCLUSION

In this work, we introduced HalluStep, a novel evaluation method designed to address the inherent volatility and lack of transparency in traditional LLM-as-a-judge metrics. By decomposing monolithic user queries into a structured set of latent, query-grounded subgoals, HalluStep shifts the evaluation paradigm from unconstrained semantic assessment to a rigorous, state-based verification process. Our experimental results across the Blocksworld and Logistics domains demonstrate that HalluStep provides a significantly more stable and reliable evaluation signal than G-Eval. Specifically, we observed that HalluStep outperforms G-Eval by providing four times more stability across judge models, higher discriminative precision through PDDL-derived subgoals, and superior scalability in complex domains like Logistics by accurately tracking state-based task fulfillment.

While current experiments establish the method's stability and discriminative precision, this remains an ongoing research effort. We are currently expanding our experimental campaign to address cross-domain generalization, conducting large-scale evaluations on legal and medical procedural tasks to assess HalluStep's robustness in specialized, high-stakes knowledge domains. Future iterations will incorporate human-in-the-loop validation via a comprehensive correlation study with expert human annotators to further validate our matching method against human intuition of logical consistency. By shifting the evaluation paradigm from holistic impressions to state-based verification, HalluStep provides a scalable foundation for measuring the reliability of AI agents in complex, multi-step environments.

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

# Appendix

.

---

**Prompt**

**Act as a Formal Logic Validator.** Your task is to perform a *State-Transition Analysis* on the provided plan. You must detect *Hallucinations* defined as:

1. **Entity Hallucination:** Using an object not defined in the Initial State.
2. **Precondition Hallucination:** Performing an action when its requirements are not met.
3. **Effect Hallucination:** Assuming a state change occurred without an action.

**Evaluation Steps:**

1. Extract the Initial State and the Goal State.
2. For EACH step $S_i$:
   – (a) Verify preconditions
   – (b) Update the *current state*
   – (c) Check for illegal actions.
3. After the final step, verify if the current world state matches the Goal State.

---

**Prompts for GEval (Step-wise Logical Grounding)**

