# OpenReview forum: "A Subgoal-Based Method for Quantifying Procedural Hallucinations in Large Language Models"
_ICLR.cc/2026/Workshop/AFAA — Submitted to AFAA 2026_

### Official Review · Reviewer_WXtQ · 2026-02-11
**Review for "A Subgoal-Based Method for Quantifying Procedural Hallucinations in Large Language Models**

**Rating:** 1
**Confidence:** 4

**Summary:**

This paper proposes HalluStep, a method for evaluating procedural hallucinations in LLMs by decomposing queries into subgoals and measuring alignment between LLM-generated plans and those subgoals. The method uses sentence embeddings for step-subgoal similarity, maximum-weight bipartite matching via the Hungarian algorithm, and derives composite scores from coverage, precision, and temporal convergence components. The authors compare HalluStep against G-Eval on Blocksworld and Logistics planning domains from PlanBench, arguing that HalluStep offers greater stability across judge models.

**Strengths:**

## 1. Relevant problem

Evaluating procedural hallucinations, where individual steps may be locally plausible but globally incoherent, is an underexplored and important problem. The distinction between under-specification (missing steps) and over-specification (hallucinated steps) is a useful conceptual framing.

## 2. Use of established planning benchmarks
Evaluating on PlanBench with PDDL-derived ground truth is a sound choice for validating a procedural evaluation metric.

**Weaknesses:**

## 1. Methods Section Contains Fundamental Errors and Incoherent Formalism

The methods section (Section 3) has serious technical problems that undermine the paper's core contribution.

 - **Equation 1 (Facility Location Objective) appears to be incorrect.** The objective maximizes $(1 - \text{sim}(c, a))$ — i.e., *dissimilarity* between candidates and the selected set. A standard facility location objective for representative selection maximizes similarity. As written, the greedy algorithm would select subgoals maximally *different* from the candidate pool. This is either a typographical error in the paper's central formulation or a fundamental misunderstanding of the objective.

 - **Subgoal salience is conceptually questionable.** Measuring cosine similarity between an atomic subgoal and an entire user query conflates two very different levels of semantic granularity, and it is unclear what meaningful signal such a score would carry.

 - **The "Precision" metric (Equation 9) is misleadingly named.** The denominator $\sum_{i=1}^{n} \max_j W_{ij}$ represents the greedy (non-one-to-one) alignment score. The ratio thus measures quality degradation from enforcing one-to-one matching constraints, not precision in any standard sense. This is never explained.

 - **The composite score (Equation 11) lacks clear interpretation.** $F_1(\text{Coverage}, \text{Precision})$ treats Coverage as recall and their redefined "Precision" as precision, but these do not have the standard precision/recall relationship. The resulting F1 is not meaningfully interpretable.

## 2. Notation and Definitions are Inconsistent, Undefined, or Unused

 - $C(t)$ in Equation 10 (Temporal Convergence) is described as the "prefix coverage curve" but never formally defined.
 - Subgoals are referred to as $G$, $C$, and $A$ at various points; the relationship between these sets ($G = A$? $G \subseteq C$?) is never made explicit. Steps are denoted as both $\pi$ and $S$, used interchangeably.
 - Subgoal salience is introduced twice under different names and notation (i.e. as $p_0(c)$ ("prior probability (or query salience)") in Equation 1 and as $q(c)$ ("Subgoal Salience") in Equation 3) without acknowledging they appear to be the same concept.

## 3. Experiments are Incomplete, Selectively Reported, and Misleadingly Described

 - **Selective discussion of favorable results.** Examining Table 1 carefully, G-Eval outperforms HalluStep in more than half of the comparisons. The paper highlights only favorable cases marked with * while not adequately discussing the many cases where HalluStep loses. A fair assessment of the method requires honest engagement with these results.
 - **The stability argument is trivially true.** HalluStep is deterministic by construction (fixed embeddings + Hungarian algorithm). Comparing its zero cross-judge variance against G-Eval's nonzero variance is not an empirical finding but a property of the method class.
 - **No ablation studies.** The method involves several design choices (embedding model, similarity threshold, subgoal granularity) but none are ablated. It is impossible to assess which components are load-bearing.


## 4. Writing Quality is Below Publication Standard

Beyond the issues in Section 3, the paper contains errors throughout. Sentence fragments, missing periods, and inconsistent capitalization appear in multiple sections. Section 4.4 contains obvious copy-paste artifacts (e.g., "A primary advantage of key advantage of HalluStep is its stability... is its stability wherease LLM as judge framwork change their scores at each run."). The paper also lacks a Limitations section.

## 5. Poor Fit with Workshop Scope

This paper addresses procedural hallucination evaluation, which is a valid research topic, but one with essentially no connection to fairness, bias, alignment, or the societal impacts that are central to the AFAA workshop. The paper does not discuss fairness in any capacity. It does not examine whether hallucination rates differ across demographic groups, whether procedural failures have differential societal impact, or how the metric relates to alignment. A planning evaluation or NLG workshop would be a far more appropriate venue.

---

### Official Review · Reviewer_HWsS · 2026-02-17
**A structured metric for procedural hallucinations**

**Rating:** 2
**Confidence:** 4

**Summary:**

The paper introduces HalluStep, a metric designed to detect procedural hallucinations in Large Language Models (LLMs), where individual reasoning steps may be valid in isolation but inconsistent or irrelevant within the overall procedure. HalluStep decomposes input queries into subgoals and evaluates whether model outputs satisfy structural milestones derived from a domain ontology. Beyond step ordering, the metric aims to identify under-specified and over-specified responses, including missing or hallucinated steps. In contrast to existing “LLM-as-judge” approaches, HalluStep is presented as interpretable and deterministic. Experiments on the PlanBench benchmarks demonstrate its robustness compared to the G-Eval baseline.

**Strengths:**

**1. Clear problem formulation.** The paper clearly defines procedural hallucinations and motivates why evaluating step-level coherence is important.

**2. HalluStep modeling.** Section 3.1 clearly defines the setup for the framework and relevant criteria for valid subgoals.

**3. Baseline comparison and statistical testing.** The comparison with the G-Eval baseline and inclusion of significance testing in Table 1 strengthen the empirical evaluation.

**Weaknesses:**

**1. Framing issues.** The abstract does not clearly distinguish procedural hallucinations from broader hallucination phenomena, which may create confusion about the scope of the paper.

**2. Methodological clarity issues.** Section 3 is difficult to follow. The transition from candidate subgoals (C, §3.3) to selected subgoals (𝒢, §3.4) to validated subgoals (G, §3.5) is not clearly explained. The selection of the candidate set size K is also unspecified in section 3.3. The definition of the prefix coverage curve C(t) (§3.6) lacks clarity. Additionally, related work states that HalluStep infers subgoals from a domain ontology, but this process is not described in the methodology.

**3. Metric design justification.** The choice of the final composite score (F1(coverage, precision) × convergence) is insufficiently justified. Alternative formulations (e.g., additive combinations) and/or ablation studies would help assess whether this design is principled. Furthermore, HalluStep is described as yielding a non-binary score, but the experiments mention that the metric classifies a plan as valid or invalid; if a threshold was used, it should be specified.

**4. Inconsistencies and contradictions.**
- Section 3.2 claims traditional metrics evaluate "responses as a whole", yet related work discusses metrics that use atomic fact decomposition or the work of Anika et al that uses procedural steps.
- In section 4.2 G-Eval judges are reportedly used with temperature 0, yet standard deviations across runs are reported in section 4.4, and G-Eval is described as stochastic in the caption of Table 1.
- HalluStep is described as judge-independent, yet in the text it is stated that performance varies across judge models (62.79 with DeepSeek v3.1 to 63.95 with Llama 3.1 405B for the same setting). Table 1 could be reformatted (one column for HalluStep, and 3 separate columns per judge for G-Eval) to improve clarity.

**5. Evaluation limitations.** The selection of target models is not explained. Fewer models are evaluated in Logistics without justification, and DeepSeek v3.2 is not used consistently across all target models. The paper mentions a closely related work (e.g., Anika et al.) but does not include it as a baseline. Finally, although the paper claims reduced computational cost, no analysis of resource usage is provided.

**6. Typos.** Some typos are present : l.088 “In contrast, Current benchmarks” → “In contrast, current benchmarks”; l.323-324 “In contrast, LLM-as-a-Judge frameworks are often inconsistent, producing different scores with each run. is its stability wherease LLM as judge framwork change their scores at each run.”, the last part should have been deleted.

**7. Missing limitations section.** Finally, the authors should have included a discussion of limitations.

---

### Official Review · Reviewer_Wecf · 2026-02-21
**A Subgoal-Based Method for Quantifying Procedural Hallucinations in Large Language Models**

**Rating:** 4
**Confidence:** 4

**Summary:**

This paper introduces HalluStep, a structured evaluation method for detecting hallucinations in procedural reasoning tasks produced by large language models. The key idea is to move away from holistic or judge based semantic evaluation and instead decompose a user query into a set of latent subgoals that define what it actually means to solve the task correctly. The generated response is then aligned against these subgoals using sentence embeddings and maximum weight bipartite matching, producing interpretable component scores that capture coverage, alignment quality, and temporal convergence. The method is evaluated on classical planning domains from PlanBench, specifically Blocksworld and Logistics, and compared against LLM as a judge approaches such as G Eval. The experiments show that HalluStep is substantially more stable across judge backbones and runs, and in many cases provides a more discriminative signal for plan correctness than stochastic judge based evaluation. The paper positions HalluStep as a deterministic and interpretable alternative for evaluating procedural hallucinations in agentic and planning oriented LLM settings

**Strengths:**

This is a strong and well thought out paper for a workshop setting. The core contribution is clear and well motivated, namely that current hallucination metrics are poorly suited for procedural and state dependent tasks. The framing around subgoals as the correct unit of evaluation makes a lot of sense and is grounded both in planning literature and cognitive models of procedural reasoning. I particularly appreciate that the method does not rely on an LLM judge at evaluation time, which directly addresses the volatility and reproducibility issues that many recent papers have pointed out but not really solved. The use of maximum weight bipartite matching to avoid double counting steps is a clean design choice and adds real interpretability. The experiments are convincing for the chosen domains, and the stability analysis across different judge models is one of the most compelling parts of the paper. For a workshop paper, the empirical section is quite thorough, and the authors do a good job explaining what each component of the metric captures and why it matters.

**Weaknesses:**

My main concern is about scope and generality rather than correctness. The method relies heavily on the availability of well defined subgoals, either derived from domain ontologies or classical planning representations. This is perfectly reasonable for Blocksworld and Logistics, but it is much less clear how HalluStep would scale to real world procedural tasks where the subgoals are ambiguous, subjective, or contested. While the paper discusses future work in legal and medical domains, at present this remains speculative. Related to this, the subgoal extraction step itself depends on embedding similarity and a facility location objective, which introduces its own inductive biases, but these are not really interrogated. Another issue is that while the paper positions itself within hallucination research, it focuses almost entirely on procedural correctness and goal satisfaction. Other forms of hallucination, such as subtle factual distortions within a correct plan, are not addressed. Finally, in the context of the AFAA workshop specifically, the connection to fairness and alignment is somewhat indirect. The paper is clearly relevant to agent evaluation and reliability, but it does not engage explicitly with fairness concerns or differential impacts across users or groups, which may limit how well it fits the workshop theme.

---

### Meta-Review · Area_Chair_9sAX · 2026-02-27

**Recommendation:** Reject
**Confidence:** 5

**Metareview:**

This paper introduces HalluStep, a structured evaluation method for detecting hallucinations in procedural reasoning tasks produced by large language models. HalluStep decomposes input queries into subgoals and evaluates whether model outputs satisfy structural milestones derived from a domain ontology. The authors compare HalluStep against G-Eval on Blocksworld and Logistics planning domains from PlanBench, arguing that HalluStep offers greater stability across judge models.

The paper studies an interesting problem, but it does not seem to be in scope for the workshop, as it does not consider fiarness. Furthermore, the reviewers identified errors and inconsistencies in the paper's key results, and one reviewer found the writing quality to be a major hurdle. Therefore, I recommend rejection at this stage of the paper. I hope the authors will take these comments to improve their paper.

---

### Decision · Program_Chairs · 2026-03-02

Reject